# A Thermomechanical Analysis of Conformal Cooling Channels in 3D Printed Plastic Injection Molds

**Suchana Akter Jahan [1],\* and Hazim El-Mounayri [2]**

[1]  School of Mechanical Engineering, Purdue University, West Lafayette, IN 47907, USA
[2]  Mechanical and Energy Engineering Department, Indiana University Purdue University Indianapolis, 723 W Michigan St, Indianapolis, IN 46202, USA; helmouna@iupui.edu
\*   Correspondence: sjahan@purdue.edu; Tel.: +1-317-954-4133

**Abstract:** Plastic injection molding is a versatile process, and a major part of the present plastic manufacturing industry. The traditional die design is limited to straight (drilled) cooling channels, which don't impart optimal thermal (or thermomechanical) performance. With the advent of additive manufacturing technology, injection molding tools with conformal cooling channels are now possible. However, optimum conformal channels based on thermomechanical performance are not found in the literature. This paper proposes a design methodology to generate optimized design configurations of such channels in plastic injection molds. The design of experiments (DOEs) technique is used to study the effect of the critical design parameters of conformal channels, as well as their cross-section geometries. In addition, designs for the "best" thermomechanical performance are identified. Finally, guidelines for selecting optimum design solutions given the plastic part thickness are provided.

**Keywords:** additive manufacturing; design of experiments; conformal cooling channels; design and analysis; design rules for additive manufacturing

---

## 1. Introduction

Our everyday life is filled with various types of plastic products. Plastic injection molding is a widely used manufacturing process that consumes a large percentage of the total amount of plastics [1]. Different complex sizes and shapes of high-quality products can be produced by this versatile process from thermoplastic and thermosetting materials with the application of heat and pressure [2]. The design of die core and cavity is very critical in the economic aspects of the injection molding business, a better quality and cost-effective product is of the utmost importance in today's competitive market.

The cooling of injection molding tooling plays a very important role in the total production cycle time of the injection molding process. The cooling time constitutes about half of the overall production cycle; hence, it is a significant portion of the entire molding process [3]. The effect of the process parameters on the polymer crystallization in plastic injection molding can be found in [4].

Injection molding is a highly used multipurpose manufacturing process for the production of plastic parts that is accepted all over the world. Traditionally, there are straight drilled holes in the solid dies that help cool the hot molten plastic inside the cavity. The cooling step is a major portion of the production cycle, resulting in a high cost of production. With the rising competition worldwide in the plastic product business, it is very important to reduce the production cost, which can be achieved by reducing the production cycle time. Implementing the design of the conformal cooling channel is a good choice for this purpose. Conformal cooling channels can improve the performance of the molds in many ways, for example; uniform and fast cooling, less warping and defects, etc. As we know that

any kind of channels could be produced using the additive manufacturing process, this technology can bring tremendous development and business benefits to the plastic injection molding industry.

The use of cooling channels that are conformal to the molding cavity can improve the control of the mold temperature and part dimensions, as reported by a group at MIT in the 1990s [5]. Xu and Sachs at MIT presented a modular, systematic approach for the design of conformal cooling channels. They divided the tool into different geometric regions and created the channel systems for each region, recognizing cooling as local to the surface of the tool. The different design parameters that were considered in the study considered in the study included the mold surface temperature, pressure drop, mold material strength, etc. [5–7]. Three-dimensional (3D) printing technology was implemented for the direct fabrication of tooling using metal powders, and improved the thermal management, dimensional control, surface finishing, and tool hardening. Stainless steel powder with a resultant tooling hardness of 25–30 Rockwell C was used in the study [8].

A study on rapid soft tooling for plastic injection molding was conducted by Ferreira and Mateus. They proposed some original approaches to integrate the advanced processing technologies, which featured composite materials chilled by conformal cooling channels in injection molding tools [9]. Meckley and Edwards presented the effect of a conformal cooling channel on reducing the cooling time and increasing part quality, in comparison with traditional straight cooling channels. They used high-density polyethylene and polycarbonate in their study, and demonstrated that the mold and melt temperature differences between the two materials illustrated the efficiency of the conformal channels [10]. The use of conformal cooling channels to both heat and cool a single injection molding tool was demonstrated in the research work by Hopkins and Dickens. This paper discussed the potential of 3D printing technology for achieving the successful production of complex geometries [11]. Yoo provided an investigation on the advantages of rapid tooling methods to build heating and cooling channels in order to enhance thermal control [12]. He also demonstrated how to seal such channels rapidly and inexpensively.

Altaf et al. provided an insight on the conformal channel fabrication method, which is not possible using traditional drilling or machining processes [13]. They presented a technique for fabricating conformal cooling channels in an aluminum-filled epoxy mold using Rapid Prototyping (RP) techniques. An investigation of the automation of the preliminary design stage to the layout design stage of the cooling system design process was presented by Li et al., who provided a configuration of straight cooling channels based on the size and shape of the plastic part design, which does not necessarily require an additive manufacturing technique [14].

There have been a series of studies in the area of design and modeling of conformal cooling channels in injection molding tooling for a long time, yet the concept of simulating the designs cannot be traced back more than 10–12 years. Since then, researchers have been using different commercially available simulation packages to analyze the tool and channel designs. Dima et al. found the best position of the runner in 2005, using Moldflow analysis in I-DEAS™ [3]. Two years later, Saifullah and Masood analyzed part cooling times using the ANSYS thermal analysis [15]. Two more years later, this research group demonstrated comparative part analysis using MPI simulation software for conventional and square section conformal cooling channels; they concluded that conformal channels render 35% less cooling time than conventional ones [16]. By incorporating a square sectioned conformal cooling channel system for injection mold dies, they provided comparative studies between conformal and traditional molds [17]. Finite element analysis using ANSYS was also presented for a mold with bimetallic conformal cooling channels. They compared the performance with a conventional mold, and provided experimental verification with two different plastic materials that were produced by a miniature injection molding machine [16]. Xu and Sachs presented a quantitative guidance for tooling design in 2009. Their proposed methodology was tested on a 3D printed benchmark tool with truss support. In their study, preliminary tests demonstrated the technical feasibility of using a solid freeform fabrication process to create low thermal inertia tools [18].

Another finite element study was presented by Sun et al. for the milled groove insert method for the cooling of plastic injection molds using a household iron plastic part [19]. This analysis was based on a cooling and thermal stress modeling technique [20]. Gloinn et al. performed finite element analysis (FEA) to determine the mold temperature using ABS polymer as the molten material and water at 20 °C as the cooling fluid [21]. Au and Yu conducted a study to investigate the thermal effects of cooling channel design on injection molding using Moldflow Plastic Insight 3.1 [22]. They proposed a novel scaffold for the design of uniform conformal cooling. A few years later in 2013, Hsu et al. identified that for cavities with irregular geometry, the distance between the cooling channels and the cavity would vary throughout the part, and would cause local heat accumulation and product defects such as sink mark, warpage, etc. They adopted a true 3D simulation technique to predict the cooling time and compare the results with traditional molds [23]. Dang and Park adopted an algorithm to calculate the temperature distribution through molds, and presented a conformal channel design with an array of baffles for obtaining uniform cooling over the entire freeform surface of the molded parts [24]. In addition to that, they provided an insight into the use of conformal cooling channels to provide a uniform cooling and reduce the cycle time for the injection molding process. U-shaped milled groove conformal channels were presented, and an optimization process was also proposed to obtain an optimal configuration of the conformal channels [25]. The comparative effect of conventional, series, parallel, and additive parallel cooling channels was studied by Khan et al. in 2014 with respect to cooling time, total cycle time, volumetric shrinkage, and temperature variance using AMI software [26]. In another study, Zink et al. pointed out the effect of limescale on cooling efficiency, also in the cases of conformal cooling channels [27].

Wang et al. presented an automatic method for designing conformal cooling circuits by establishing a relationship between the conformal cooling and the shape of the plastic body [28]. Choi et al. established a higher degree of freedom in the design of conformal cooling channels with the application of additive manufacturing and concentrate on a branching law principle to improve the cooling efficiency in injection molds [29]. To create the design of conformal cooling channels, they used the Voronoi diagram algorithm and the binary branching algorithm. A similar technique was also adopted by Park and Pham. They designed cooling channels for individual surfaces, and then combined them to form an overall conformal cooling channel system for the entire part [30]. Two years later, they designed conformal cooling channels for an automotive part using the algorithm that they provided in their previous work. In that study, they conducted an optimization to minimize the cooling time with boundaries ensuring a realistic design for the cooling system [31]. Wang et al. introduced an approach to generate spiral channels for conformal cooling. Using boundary distance maps, their algorithm could generate evenly distributed spiral channels in the injection mold [32].

In 2011, a design methodology called visibility-based cooling channel generation was presented by Au and Yu for an automatic preliminary cooling channel design. This was more of a geometric and theoretical method, rather than intended for a practical scenario [33]. Subsequently, this research group provided a cooling channel distance modification based on adjustments to the direction and amount in 2014. Also, a simulation technique using MoldFlow Plastic Insight software was adopted to demonstrate the feasibility of their proposed method [34]. Agazzi et al. proposed a new methodology called "Morpho Cooling" for the design of cooing channels in the injection mold. This method provided better results in cooling in terms of the higher uniformity of temperature distribution and lesser part warpage [35].

It is evident that there have been a lot of studies about the analysis of conformal cooling channels, yet, the number of studies dedicated to the design parameters of conformal channels for various kinds of part designs is very limited. In the mold and tooling industry, most of the designs are done based on the designers' experiences. Also, any kind of mix and match between the design parameters, cross-section size, and respective experimental analyses is pretty rare, according to the author's knowledge. Yet, some preliminary information could be gathered from the literature that act as a basis for further research on this project. For example, a simple relationship between four parameters for the

design of conformal cooling channels using additive manufacturing is found from Mayer [36]. The data is adapted from their study, and this relationship is shown in Table 1. Some studies show that the use of different cross-sections for channels other than circular might provide better cooling efficiency.

**Table 1.** Correlation amongst design parameters of conformal cooling channels [37].

| Wall Thickness of Molded Part (mm) | Channel Diameter, D (mm) | Pitch Distance, P (mm) | Channel Centerline to Mold Wall Distance, L (mm) |
|---|---|---|---|
| 0–2 | 4–8 | 2D–3D | 1.5D–2D |
| 2–4 | 8–12 | 2D–3D | 1.5D–2D |
| 4–8 | 12–14 | 2D–3D | 1.5D–2D |

From the literature review, it is evident that there have been a number of studies in the field of design that provide an analysis of conformal cooling channels in injection molds. Some of the studies mostly discussed the process of designing conformal channels and the design parameters. Some other papers concentrated on the production process of such conformal cooling channels. Again, some provided results of numerical analyses of injection molds. Yet, no study has yet identified how to incorporate conformal cooling channels into injection molds in order to most effectively provide both thermal and structural performance. This is the principal motivating objective of this research effort. The effort of identifying a design technique for optimal conformal cooling channels in injection molds has been initiated in recent years by Jahan and Wu et al. [37–43]. This current paper provides a guideline for the mold designers to design their injection molds with the conformal cooling channels that would enable them to obtain the most benefit in business. A number of sets of design of experiments (DOEs) are presented, where thermomechanical analysis is performed on all the design cases in all of the DOEs, and the outcomes are compared to finally identify the most effective design configuration for the conformal cooling channels. This optimization is expected to provide useful insight for mold designers in the plastic injection molding industry.

## 2. Materials and Methods

We propose a methodology for creating conformal cooling channels in injection molds that result in better performance regarding fast and uniform cooling, structural stability, reduced cycle time, and improved part quality. By applying additive manufacturing, we can create cooling channels of any size and shape inside the mold core and cavity, which is not possible using traditional manufacturing processes.

In the recent publication by the current authors [37,42], numerical models to analyze the thermal and structural behavior of plastic injection molds were developed and validated. In that study, the authors generated an optimal design of conformal cooling channels for a specific size and shape of a plastic part. Moreover, a cooling channel with a rectangular cross-section provided shorter cooling time. In this paper, the authors propose a general design methodology for conformal cooling channels. Here, three different values of the thickness of plastic parts are considered, and a thermomechanical optimization is achieved through a design of experiments approach. The design of the plastic part to be manufactured by injection molding affects the design of the cooling system of the mold. For example, the higher the thickness of the plastic part, the larger the cooling channel diameter should be. Mayer [36] provided a guideline showing the correlations between the design variables of conformal cooling channels. In the current study, the design variables are kept independent in all of the DOEs, and are considered within a range to allow for a comprehensive design solution.

### 2.1. Design of Experiments for Optimized Conformal Channels

A design of experiments approach is used to guide the design of conformal cooling channels. Typically, the thickness of molded plastic parts varies within a range of 0–6 mm. Different shapes, sizes, and thicknesses of the part would require different configurations of optimized conformal

cooling channels. As such, only one basic shape of plastic parts, namely cylindrical, is considered. Three different thicknesses are modeled: one mm, 3.5 mm, and six mm. Thus, three sets of DOEs are prepared: DOE-1 for the thickness of one mm; DOE-2 for the thickness of 3.5 mm; and DOE-3 for the thickness of six mm.

The information regarding the part thickness provides the basic outline for channel design parameters such as diameter (in circular channels), pitch distance, channel centerline to mold wall distance, etc. The range of analysis for each DOE has been determined from the literature and the general rule of thumb of mold designers [36]. After deciding on the design variables (pitch, wall to wall distance, etc.), the channel cross-section is selected. This research group found that a rectangular cross-section of conformal cooling channels provides effective thermal performance in injection molds. As a result, all of the design cases in the above-mentioned DOEs are created with rectangular-shaped cooling channels. The DOE parameter details are mentioned in Table 2. The design cases are designated as 1.1, 1.2 . . . , 2.1 . . . , 3.1, 3.2, . . . 3.18.

**Table 2.** Case setup variables in design of experiments (DOEs) 1, 2 and 3, in which the thickness of the plastic parts are set at one mm, 3.5 mm, and six mm, respectively.

| DOE-1 | | | | DOE-2 | | | | DOE-3 | | | |
|---|---|---|---|---|---|---|---|---|---|---|---|
| Case Number | X-Section (mm× mm) | P (mm) | L (mm) | Case Number | X-Section (mm× mm) | P (mm) | L (mm) | Case Number | X-Section (mm× mm) | P (mm) | L (mm) |
| 1.1 | $3.8 \times 2.5$ | 8 | 6 | 2.1 | $7.5 \times 5$ | 16 | 12 | 3.1 | $11.3 \times 7.5$ | 24 | 18 |
| 1.2 | $3.8 \times 2.5$ | 8 | 16 | 2.2 | $7.5 \times 5$ | 16 | 24 | 3.2 | $11.3 \times 7.5$ | 24 | 28 |
| 1.3 | $3.8 \times 2.5$ | 16 | 6 | 2.3 | $7.5 \times 5$ | 26 | 12 | 3.3 | $11.3 \times 7.5$ | 33 | 18 |
| 1.4 | $3.8 \times 2.5$ | 16 | 16 | 2.4 | $7.5 \times 5$ | 26 | 24 | 3.4 | $11.3 \times 7.5$ | 33 | 28 |
| 1.5 | $3.8 \times 2.5$ | 24 | 6 | 2.5 | $7.5 \times 5$ | 36 | 12 | 3.5 | $11.3 \times 7.5$ | 42 | 18 |
| 1.6 | $3.8 \times 2.5$ | 24 | 16 | 2.6 | $7.5 \times 5$ | 36 | 24 | 3.6 | $11.3 \times 7.5$ | 42 | 28 |
| 1.7 | $5.6 \times 3.8$ | 8 | 6 | 2.7 | $9.4 \times 6.3$ | 16 | 12 | 3.7 | $12.3 \times 8.2$ | 24 | 18 |
| 1.8 | $5.6 \times 3.8$ | 8 | 16 | 2.8 | $9.4 \times 6.3$ | 16 | 24 | 3.8 | $12.3 \times 8.2$ | 24 | 28 |
| 1.9 | $5.6 \times 3.8$ | 16 | 6 | 2.9 | $9.4 \times 6.3$ | 26 | 12 | 3.9 | $12.3 \times 8.2$ | 33 | 18 |
| 1.10 | $5.6 \times 3.8$ | 16 | 16 | 2.10 | $9.4 \times 6.3$ | 26 | 24 | 3.10 | $12.3 \times 8.2$ | 33 | 28 |
| 1.11 | $5.6 \times 3.8$ | 24 | 6 | 2.11 | $9.4 \times 6.3$ | 36 | 12 | 3.11 | $12.3 \times 8.2$ | 42 | 18 |
| 1.12 | $5.6 \times 3.8$ | 24 | 16 | 2.12 | $9.4 \times 6.3$ | 36 | 24 | 3.12 | $12.3 \times 8.2$ | 42 | 28 |
| 1.13 | $7.5 \times 5$ | 8 | 6 | 2.13 | $11.3 \times 7.5$ | 16 | 12 | 3.13 | $13.2 \times 8.8$ | 24 | 18 |
| 1.14 | $7.5 \times 5$ | 8 | 16 | 2.14 | $11.3 \times 7.5$ | 16 | 24 | 3.14 | $13.2 \times 8.8$ | 24 | 28 |
| 1.15 | $7.5 \times 5$ | 16 | 6 | 2.15 | $11.3 \times 7.5$ | 26 | 12 | 3.15 | $13.2 \times 8.8$ | 33 | 18 |
| 1.16 | $7.5 \times 5$ | 16 | 16 | 2.16 | $11.3 \times 7.5$ | 26 | 24 | 3.16 | $13.2 \times 8.8$ | 33 | 28 |
| 1.17 | $7.5 \times 5$ | 24 | 6 | 2.17 | $11.3 \times 7.5$ | 36 | 12 | 3.17 | $13.2 \times 8.8$ | 42 | 18 |
| 1.18 | $7.5 \times 5$ | 24 | 16 | 2.18 | $11.3 \times 7.5$ | 36 | 24 | 3.18 | $13.2 \times 8.8$ | 42 | 28 |

For DOE-1, the plastic part thickness is one mm. According to the guidelines in Table 1, the channel diameter should be in the range of four to eight mm. The perimeter of such channels would be 12.6 mm to 25.13 mm. Keeping the perimeter the same, the circular channels are converted into rectangular ones, and their cross-sectional dimensions are calculated. For example, the circular channel with a diameter of four-mm was converted to a rectangular channel with a cross-section of 3.8 mm × 2.5 mm, and the channel with a diameter of eight mm is converted into a 7.5 mm × 5 mm section channel. There are three design variables in each DOE 1, 2 and 3. These are the channel cross-section (a × b), pitch distance (P), and mold wall to channel centerline distance (L). The first two variables have three levels of design, and the third one has two levels of design. As a result, DOE-1 has 3 × 3 × 2 = 18 design cases (Table 2, column 1). Similarly, DOE-2 has 18 cases (Table 1, column 5), and DOE-3 also has 18 design cases (Table 2, column 9). The channel design parameters, such as pitch distance and cross-section dimensions, are the same for both the cavity and core in a single design case. In all of the DOEs mentioned here (DOE 1, 2 and 3), as well as the other ones (DOE 4, 5, 6, 7, 8 and 9) discussed in the latter part of the paper, a full factorial method is used to formulate the design of experiments.

## 2.2. Thermomechanical Optimization

In previous studies by this research group [37–45], the numerical analysis and design decisions regarding the performance of conformal cooling channels in injection molds were conducted with the single consideration of fast cooling. In addition, static structural analysis was conducted to ensure

the structural stability of the mold cores and cavities with conformal channels. Both thermal and structural analyses were performed on all of the design case studies, and a trade-off between thermal and structural performance was conducted to find the best design solution.

Theoretically, a solid mold with a single straight-drilled channel is structurally stable, and can withstand large amount of stress compared to a mold with conformal channels due to the higher void space within the body. The mold is made of structural steel (density: 7850 kg/m$^3$, thermal conductivity 60.5 W/m-K, specific heat 60.5 J/kg-K, yield strength 430 MPa). On the other hand, a mold with conformal channels can cool off the molten plastic quickly, as well as be uniformly compared to the traditional mold due to the presence of conformal channels at the vicinity of the plastic part. The plastic is polypropylene (density: 830 kg/m$^3$, thermal conductivity 0.14 W/m-K, specific heat 1900 J/kg-K). In the thermal analysis, the initial condition was set for coolant inlet at 22 °C, and the molten plastic was set at 168 °C. The plastic ejection temperature was 50 °C. For the mechanical or static structural analysis, the clamping force was 110 ton, which was applied to the top and bottom surface of the mold. In addition to that, an injection pressure of 131 MPa was applied to the heating surface. The structural analysis predicted the deformation and distribution of von Mises stress on the mold body. If the maximum von Mises stress is below the acceptable limit of yield stress of the mold material, the mold is considered to be structurally stable and functional. The design cases as mentioned in Table 2 are analyzed for both the thermal and structural behavior using the simulation technique in the published research of the author [38,42], and an optimization method is conducted for each DOE to obtain the best design scenario. The optimization problem statements for DOE-1, DOE-2, and DOE-3 are summarized in Table 3.

**Table 3.** Optimization problem statements for DOE-1, DOE-2, and DOE-3.

|  | DOE-1 | DOE-2 | DOE-3 |
|---|---|---|---|
| Objective function | 1. Minimize cooling time | 1. Minimize cooling time | 1. Minimize cooling time |
|  | 2. Minimize max von Mises stress | 2. Minimize max von Mises stress | 2. Minimize max von Mises stress |
| Design Variables | ● Channel perimeter (C) | ● Channel perimeter (C) | ● Channel perimeter (C) |
|  | ● Pitch (P) | ● Pitch (P) | ● Pitch (P) |
|  | ● Mold wall to channel centerline distance (L) | ● Mold wall to channel centerline distance (L) | ● Mold wall to channel centerline distance (L) |
| Constraints | ● Cooling time <28.04 s | ● Cooling time <28.32 s | ● Cooling time <35.55 s |
|  | ● Maximum von Mises stress <215 MPa | ● Maximum von Mises stress <215 MPa | ● Maximum von Mises stress <215 MPa |
|  | ● 12 mm< C <25 mm | ● 25 mm< C <38 mm | ● 382 mm< C <44 mm |
|  | ● 8 mm< P <24 mm | ● 16 mm< P <36 mm | ● 24 mm< P <42 mm |
|  | ● 16 mm< L <26 mm | ● 12 mm< L <24 mm | ● 18 mm< L <28 mm |

*2.3. Design Optimization of Cooling Channels for Different Shape of Plastic Parts*

The previously mentioned DOEs (DOE 1, 2 and 3) were designed considering that the final plastic part is only cylindrical in shape. However, in the real world, there are a number of various sizes and shapes of plastic products. To study the effect of this variation in shape and how it affects the optimal shape of the conformal cooling channels, six new sets of DOEs are prepared (DOE 4, 5, 6, 7, 8 and 9). Six different plastic part designs are considered here. The CAD models of these plastic parts are shown in Figure 1. There are two basic part shapes, each with three different thicknesses. Hence, six different design of experiments sets are created. Table 4 shows the DOE set with the plastic part dimension.

The design of the conformal cooling channels for these DOEs are based on the results obtained from DOE 1, 2 and 3. The best possible design solutions found from DOE 1, 2 and 3 are used as design baselines for the ones in DOE 4, 5, 6, 7, 8 and 9. This is explained in detail in the later sections.

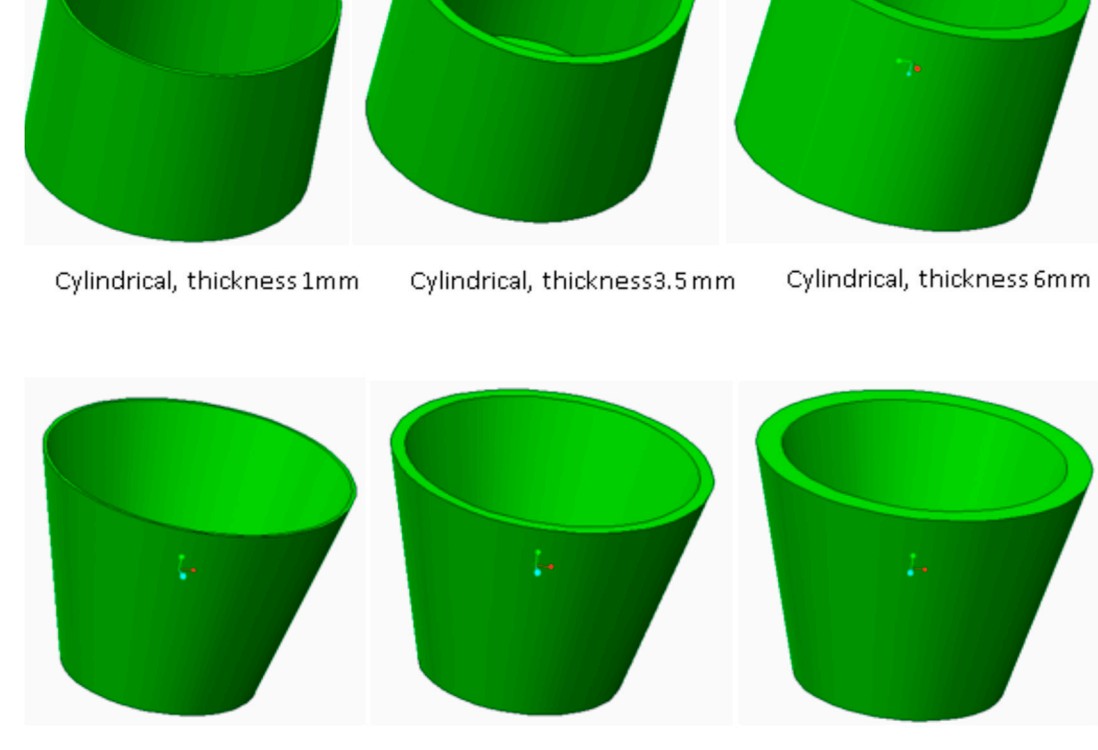

**Figure 1.** CAD models of six different plastic parts.

**Table 4.** Plastic part dimensions for DOE 4, 5, 6, 7, 8 and 9.

| DOE Number | Shape of Plastic Part | Part Thickness (mm) | Part Height (mm) | Part Larger Diameter (mm) | Part Smaller Diameter (mm) |
|---|---|---|---|---|---|
| DOE-4 | cylindrical | 1 | 60 | 80 | 80 |
| DOE-5 | cylindrical | 3.5 | 60 | 80 | 80 |
| DOE-6 | cylindrical | 6 | 60 | 80 | 80 |
| DOE-7 | conical | 1 | 60 | 80 | 50 |
| DOE-8 | conical | 3.5 | 60 | 80 | 50 |
| DOE-9 | conical | 6 | 60 | 80 | 50 |

## 3. Results and Discussions

### 3.1. Findings from DOE 1, 2 and 3

Table 5 shows the thermal and structural results obtained from DOE 1, 2 and 3. The results are in terms of cooling time and maximum von Mises stress at the time of ejection. The objective is to obtain the minimum cooling time and minimum stress, as previously mentioned in Table 3.

In Table 5 column 2, it is seen that for DOE-1, the minimum cooling time is 14.32 s in case 1.1, whereas the minimum value of the maximum von Mises stress occurs in case 1.10 (column 3, row 12), which is 107 MPa. Similar phenomena are observed for DOE-2 and DOE-3, too. In DOE-2 (Table 5 column 5), the minimum cooling time, i.e., 21.39 s, occurs in case 2.7, whereas the minimum stress 161 MPa occurs in case 2.5 (column 6). In DOE-3, case 3.13 and case 3.6 have the best thermal (27.47 s) and best structural (162 MPa) results, as found in column 8 and 9, respectively. Hence, a trade-off is necessary in order to define the best design solution.

**Table 5.** DOE 1, 2 and 3 thermal and structural results.

| DOE-1 | | | DOE-2 | | | DOE-3 | | |
|---|---|---|---|---|---|---|---|---|
| Case Number | Cooling Time (s) | Max von Mises Stress (MPa) | Case Number | Cooling Time (s) | Max von Mises Stress (MPa) | Case Number | Cooling Time (s) | Max von Mises Stress (MPa) |
| **1.1** | 14.32 | 159 | **2.1** | 24.01 | 183 | **3.1** | 28.78 | 179 |
| **1.2** | 23.83 | 155 | **2.2** | 34.59 | 176 | **3.2** | 31.91 | 180 |
| **1.3** | 14.68 | 157 | **2.3** | 28.57 | 169 | **3.3** | 32.52 | 173 |
| **1.4** | 35.97 | 206 | **2.4** | 39.22 | 164 | **3.4** | 35.39 | 170 |
| **1.5** | 26.07 | 188 | **2.5** | 32.63 | 161 | **3.5** | 35.04 | 172 |
| **1.6** | 33.36 | 156 | **2.6** | 42.87 | 162 | **3.6** | 37.15 | 162 |
| **1.7** | 17.97 | 165 | **2.7** | 22.37 | 203 | **3.7** | 28.35 | 197 |
| **1.8** | 20.08 | 299 | **2.8** | 32.59 | 196 | **3.8** | 30.78 | 174 |
| **1.9** | 17.53 | 159 | **2.9** | 26.55 | 166 | **3.9** | 36.6 | 184 |
| **1.10** | 25.22 | 107 | **2.10** | 37.33 | 163 | **3.10** | 34.24 | 168 |
| **1.11** | 23.04 | 153 | **2.11** | 31.37 | 215 | **3.11** | 33.99 | 174 |
| **1.12** | 29.64 | 161 | **2.12** | 40.71 | 168 | **3.12** | 36.35 | 171 |
| **1.13** | N/A | N/A | **2.13** | 23.58 | 244 | **3.13** | 27.47 | 176 |
| **1.14** | N/A | N/A | **2.14** | 33.45 | 222 | **3.14** | 30.52 | 189 |
| **1.15** | 16.49 | 174 | **2.15** | 26.75 | 171 | **3.15** | 30.75 | 186 |
| **1.16** | 23.88 | 204 | **2.16** | 36.03 | 169 | **3.16** | 33.35 | 177 |
| **1.17** | 21.21 | 144 | **2.17** | 29.98 | 165 | **3.17** | 33.45 | 177 |
| **1.18** | 28.1 | 169 | **2.18** | 38.57 | 167 | **3.18** | 35.65 | 175 |

It is notable that the value of the minimum cooling time increases from DOE-1 to DOE-3. This is an expected behavior, as the thickness of the plastic part also increases. Although the cooling time increases, in each case, they show better results than their respective traditional mold design scenario. This comparison is conducted by creating three traditional mold designs with straight drilled cooling channels and analyzing their thermal and structural performance. These cases are named as 1-conventional, 2-conventional, and 3-conventional. These design cases are created for plastic parts with thicknesses of one mm, 3.5 mm, and six mm, respectively, in order to be comparable with their conformal design cases. Table 6 shows the cooling times for these conventional cases along with the respective conformal design cases, which indicate positive improvement with the application of conformal cooling channels in all of the cases.

**Table 6.** Comparative thermal and structural results from DOE 1, 2 and 3 with conformal and conventional cooling channels.

| DOE Number | Conformal Cooling Time (s) | Conventional Cooling Time (s) | Conformal Max. von Mises Stress (MPa) | Conventional Max. von Mises Stress (MPa) |
|---|---|---|---|---|
| DOE-1 | 14.32 | 28.04 | 159 | 153 |
| DOE-2 | 22.37 | 28.32 | 203 | 167 |
| DOE-3 | 27.47 | 35.55 | 176 | 168 |

The thermal analysis results for the design cases mentioned in Table 6 are shown in Figure 2. It shows the temperature distribution of the respective plastic parts at the time of ejection. The selected top three design cases are listed in Table 7. This selection is based on a trade-off between the best thermal results and best structural results in DOE 1, 2 and 3.

**Table 7.** Selected top three optimal design cases for DOE 1, 2 and 3.

| DOE Number | Selected Cases | | |
|---|---|---|---|
| DOE-1 | 1.1 | 1.3 | 1.9 |
| DOE-2 | 2.1 | 2.9 | 2.17 |
| DOE-3 | 3.1 | 3.8 | 3.13 |

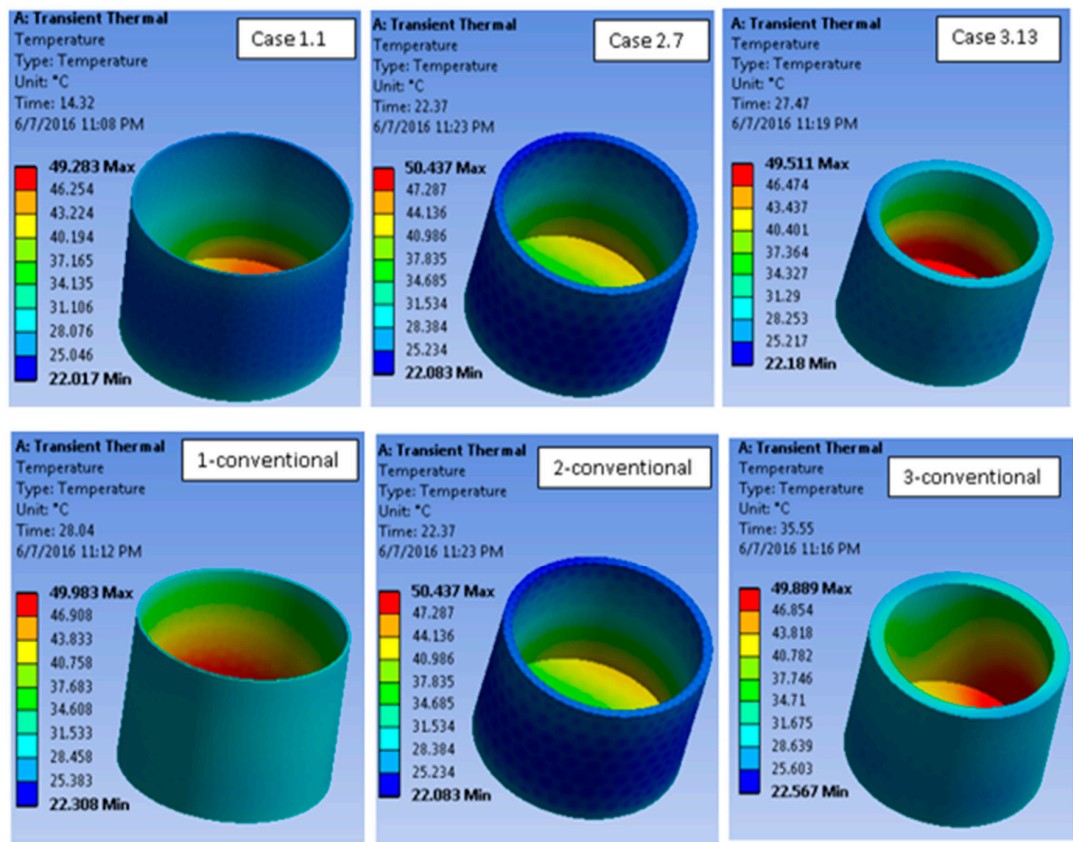

**Figure 2.** Temperature distribution on plastic parts for comparable conformal and conventional cases in DOE 1, 2 and 3.

Figure 3 shows the variation of cooling time and stress distribution with pitch distance (P) and the mold-to-channel centerline distance (L) for DOE-1. Figures 4 and 5 show similar trends for DOE-2 and DOE-3 (i.e., increase in cooling time with increase in P value). The same effect is observed for L. From these figures, it can be noted that with the increase in part thickness, the variation in cooling times decreases. Hence, the thicker the plastic part, the more critical the need for an improved conformal cooling channel design. When the plastic parts are thicker, the temperature is higher in the inner surface than the outer surface, and it is more important to keep the distance of the channels to the mold wall smaller in the core than the cavity.

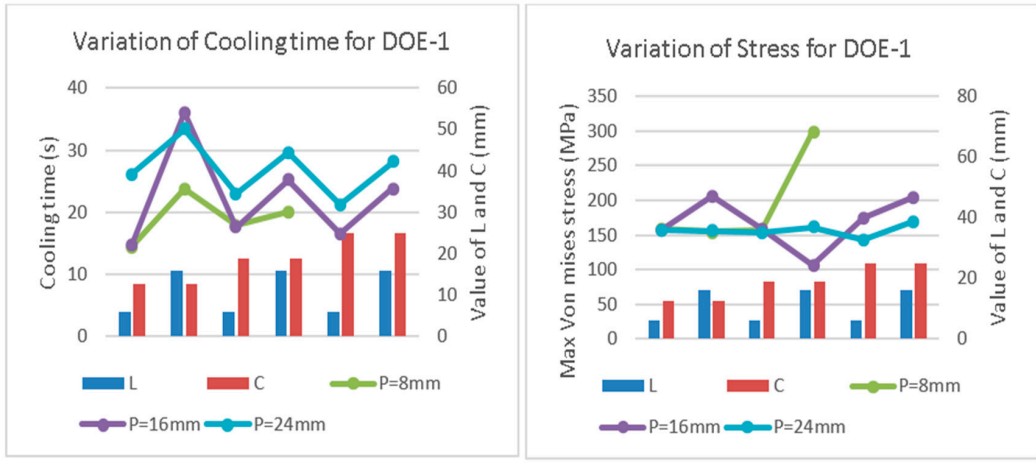

**Figure 3.** Trend of cooling time and maximum von Mises stress variation in DOE-1.

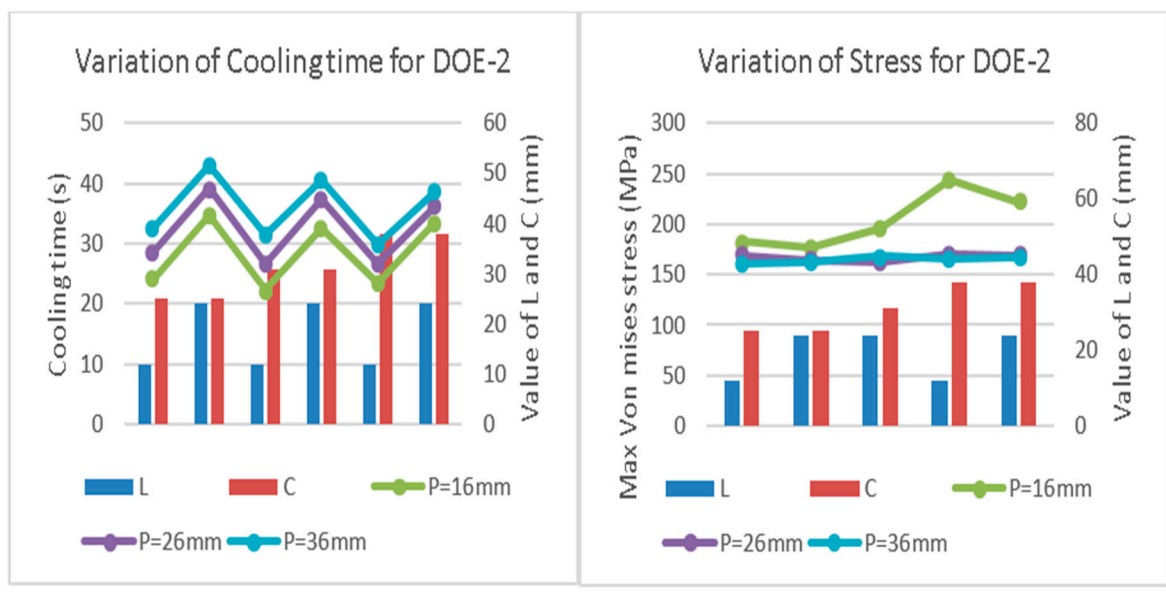

**Figure 4.** Trend of cooling time and maximum von Mises stress variation in DOE-2.

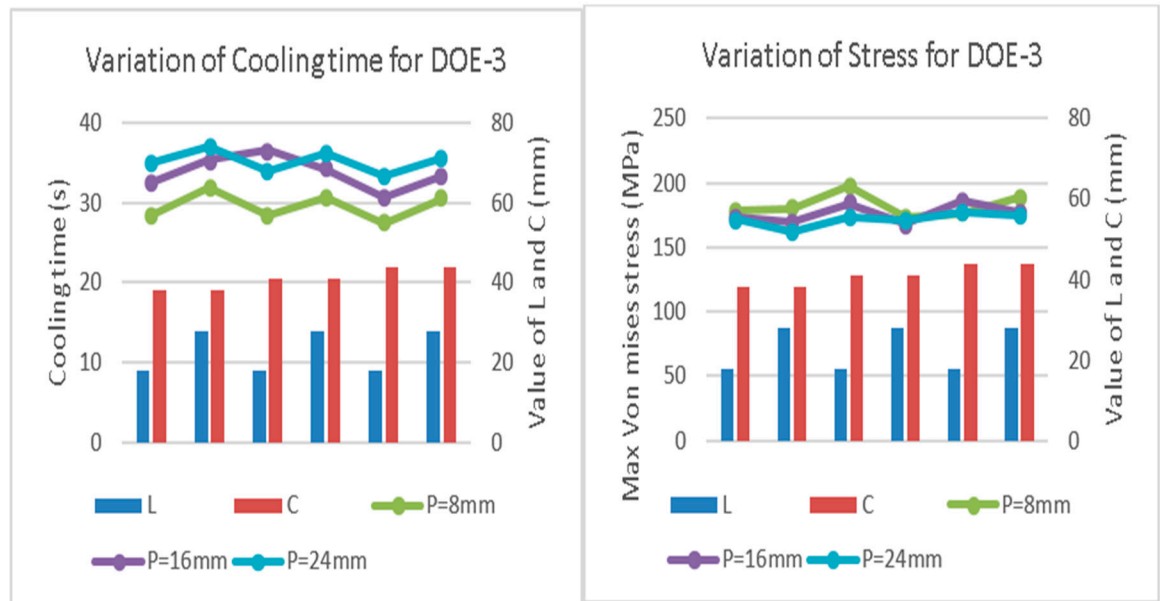

**Figure 5.** Trend of cooling time and maximum von Mises stress variation in DOE-3.

Analyzing the thermal and structural results, it is noted that although the cooling times vary significantly among the cases in each DOE set, the value of the maximum von Mises stress is relatively constant. Those values are much smaller than the yield strength. Thus, the thermal performance is emphasized to determine optimum designs. In DOE 1, 2 and 3, considering the cooling times, the variances are 38.45 s, 35.72 s, and 8.27 s respectively, while the variances are 1576.56 MPa, 563.89 MPa, and 63.43 MPa, considering the von Mises stress. With the optimal designs obtained in the previous section, the study is further extended by considering the cylindrical and conical designs of plastic parts, which are the most commonly used in the injection molding industry. The top three optimized designs of cooling channels (from Table 7) for each thickness are chosen to set up new sets of DOEs. In terms of the channel cross-section, circular, square, and rectangular designs provide comparatively better thermal performance according to a previous study [38]. These three different channel cross-sections are incorporated here.

### 3.2. Setting up DOE 4, 5, 6, 7, 8 and 9 Using Results of DOE 1, 2 and 3

As mentioned in Section 2.3, the setup of DOEs 4, 5, 6, 7, 8, and 9 is completely dependent upon the results of DOE 1, 2 and 3. As we have discussed the results of the first three DOEs already, we can now explain how the DOE 4, 5, 6, 7, 8 and 9 are created. Let's consider Table 7. It contains the optimally selected top results of DOE 1, 2 and 3. In the first row, it shows the top three design cases for DOE-1, which are case 1.1, 1.3, and 1.9. If we go back to Table 2 to understand what the channel design in case 1.1 actually is, we can see that the channel cross-section is 3.8 mm × 2.5 mm, P is 8 mm, and L is 6 mm. Now, to prepare DOE-4, we would like to keep the values of P (pitch) and L exactly the same, and change the shape of the channel cross-section. We will keep the perimeter (circumference) the same and convert the channel design into circular and square-shaped cross-sections. The resulting channels are a circular section with a diameter of four mm and a square section channel of 3.1 mm × 3.1 mm. The values of P and L are the same in all three cases. In this manner, one design case selected in Table 7 provides three different configurations (circular, rectangular, square cross-section) of conformal cooling channel designs. Thus, the nine design cases mentioned in Table 7 provide 9 × 3 = 27 types of channel designs.

These 27 design configurations of conformal cooling channels are incorporated for cylindrical and conical shapes of plastic parts, as earlier mentioned in Section 2.3. Thus, we will have nine design cases the of DOE-4, nine for DOE-5, and so on. DOE 4, 5 and 6 will have a cylindrical plastic part, and DOE 7, 8 and 9 will have a conical plastic part.

Table 8 shows the channel dimensions of these 27 conformal channel designs for cylindrical-shaped plastic parts. These 27 designs can be repeated in injection molds for conical shape plastic parts, which are assigned in DOE 7, 8 and 9. The only difference between the DOE 4, 5 and 6 and DOE 7, 8 and 9 cases is the profile of the channels. For the first case (cylindrical), the channel spiral shape is cylindrical, while the conical-shaped plastic part will have conical-shaped spirals. Hence, the channel dimensions for DOE-4 (Table 8, columns 2, 3, and 4) are exactly the same as those for DOE-7. Similarly, DOE-5 matches with DOE-8, and DOE-6 matches with DOE-9. Figure 6 illustrates the shapes of the conformal channels for cylindrical and conical parts.

**Table 8.** Channel dimensions for DOE 4 (plastic part thickness: one mm), DOE-5 (plastic part thickness: 3.5 mm) and DOE-6 (plastic part thickness: six mm).

| DOE-4 | | | | DOE-5 | | | | DOE-6 | | | |
|---|---|---|---|---|---|---|---|---|---|---|---|
| Case Number | X-Section (mm) | P (mm) | L (mm) | Case Number | X-Section (mm) | P (mm) | L (mm) | Case Number | X-Section (mm) | P (mm) | L (mm) |
| **4.1** | D = 4 | 8 | 6 | **5.1** | D = 8 | 16 | 12 | **6.1** | D = 12 | 24 | 18 |
| **4.2** | 3.8 × 2.5 | 8 | 6 | **5.2** | 7.5 × 5 | 16 | 12 | **6.2** | 11.3 × 7.5 | 24 | 18 |
| **4.3** | 3.1 × 3.1 | 8 | 6 | **5.3** | 6.3 × 6.3 | 16 | 12 | **6.3** | 9.4 × 9.4 | 24 | 18 |
| **4.4** | D = 4 | 16 | 6 | **5.4** | D = 10 | 26 | 12 | **6.4** | D = 13 | 28 | 28 |
| **4.5** | 3.8 × 2.5 | 16 | 6 | **5.5** | 9.4 × 6.3 | 26 | 12 | **6.5** | 12.3 × 8.2 | 28 | 28 |
| **4.6** | 3.1 × 3.1 | 16 | 6 | **5.6** | 7.9 × 7.9 | 26 | 12 | **6.6** | 10.2 × 10.2 | 28 | 28 |
| **4.7** | D = 6 | 16 | 6 | **5.7** | D = 12 | 36 | 12 | **6.7** | D = 14 | 24 | 18 |
| **4.8** | 5.6 × 3.8 | 16 | 6 | **5.8** | 11.3 × 7.5 | 36 | 12 | **6.8** | 13.2 × 8.8 | 24 | 18 |
| **4.9** | 4.7 × 4.7 | 16 | 6 | **5.9** | 9.4 × 9.4 | 36 | 12 | **6.9** | 10.9 × 10.9 | 24 | 18 |

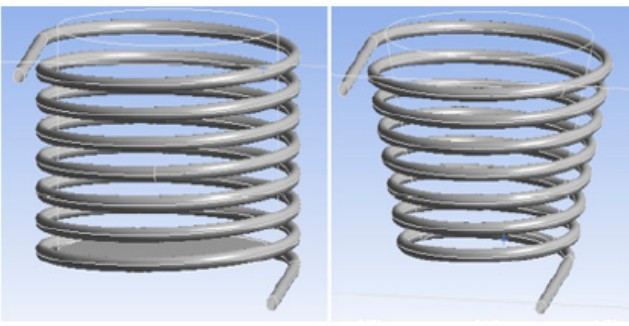

**Figure 6.** Generic cylindrical and conical configuration of conformal cooling channels.

Applying the above-mentioned channel design techniques, six design of experiment sets (DOE 4, 5, 6, 7, 8 and 9) are created and analyzed for their thermal performance. Structural analysis is not conducted for the cases of DOE 4, 5, 6, 7, 8 and 9, as the structural stability has already been checked through simulation.

### 3.3. Findings from DOE1, 2 and 3

Each of the six experimental set designs for DOE 4, 5, 6, 7, 8 and 9 has nine cases. The thermal analysis results in terms of the cooling times of these cases are shown in Table 9. The first six columns are for cylindrical cases, and the later ones are for conical cases.

**Table 9.** Thermal analysis results (cooling times) for DOE 4, 5, 6, 7, 8 and 9

| Cylindrical Part | | | | | | Conical Part | | | | | |
|---|---|---|---|---|---|---|---|---|---|---|---|
| Case Number | Cooling Time (s) | Case Number | Cooling Time (s) | Case Number | Cooling Time (s) | Case Number | Cooling Time (s) | Case Number | Cooling Time (s) | Case Number | Cooling Time (s) |
| **4.1** | 13.51 | **5.1** | 28.82 | **6.1** | 27.65 | **7.1** | 12.7 | **8.1** | 22.63 | **9.1** | 17.63 |
| **4.2** | 14.32 | **5.2** | 24.01 | **6.2** | 28.48 | **7.2** | 12.84 | **8.2** | 19.84 | **9.2** | 17.87 |
| **4.3** | 13.87 | **5.3** | 23.73 | **6.3** | 28.72 | **7.3** | 12.75 | **8.3** | 19.65 | **9.3** | 20.05 |
| **4.4** | 20.03 | **5.4** | 27.50 | **6.4** | 30.35 | **7.4** | 18.03 | **8.4** | 21.48 | **9.4** | 18.13 |
| **4.5** | 14.68 | **5.5** | 26.55 | **6.5** | 30.78 | **7.5** | 18.27 | **8.5** | 21.39 | **9.5** | 17.74 |
| **4.6** | 20.39 | **5.6** | 27.45 | **6.6** | 30.16 | **7.6** | 18.25 | **8.6** | 20.85 | **9.6** | 17.50 |
| **4.7** | 17.46 | **5.7** | 29.30 | **6.7** | 26.89 | **7.7** | 15.94 | **8.7** | 21.02 | **9.7** | 15.96 |
| **4.8** | 17.53 | **5.8** | 29.98 | **6.8** | 27.47 | **7.8** | 16.21 | **8.8** | 22.63 | **9.8** | 16.51 |
| **4.9** | 17.86 | **5.9** | 30.28 | **6.9** | 27.16 | **7.9** | 16.04 | **8.9** | 22.63 | **9.9** | 16.37 |

From Table 9, we can see that when all the other design variables are kept unchanged, the cooling time does vary if the cross-section geometry changes. This means that the effect of the cross-section is important when designing conformal cooling channels for injection molds. From the results, it is also be seen that no single cross-section geometry provides the best performance amongst all of the scenarios.

For DOE-4, the minimum cooling time obtained is 13.51 s, which occurs in case 4.1. This is a circular-shaped conformal channel with D = four mm, P = eight mm, and L = six mm. The design cases 4.2 and 4.3 have the exact same P and L values, but rectangular and square sections provide slightly higher cooling times of 14.32 s and 13.87 s, respectively. The other design cases in DOE-4 provide cooling times in the range of 14 s to 20 s. It is reasonable to mention here that this DOE-4 is only for cylindrical shapes that have one-mm thick plastic parts only, whereas DOE-5 and DOE-6 have 3.5-mm and six-mm plastic parts, respectively. As mentioned earlier, a cylindrical-shaped plastic part with one-mm thickness is cooled down to its ejection temperature at around 28 s. Hence, all of the design cases in DOE-4 (4.1–4.9) are acceptable design solutions, while a mold designer needs to do design one. It is possible for a mold designer to choose and even mix and match from any of these cases if there are some design constraints that do not permit him to choose the most effective one (case 4.1 in this scenario).

For DOE-5, the best result, 23.73 s, is achieved in case 5.3, which is a square section channel (6.3 mm × 6.3 mm), with P = 16 mm and L = 12 mm. The similar design cases (same P and L as case 5.3) provide higher cooling times. Also, the other design cases in DOE-5 provide cooling times between 25–30 s. Again, for DOE-6 (six-mm thickness plastic part), the best solution comes in case 6.7, which is 26.89 s. The channel dimensions in this case are D = 14 mm, P = 24 mm, and L = 18 mm. The overall cooling time range in DOE-6 is 27 s–30 s. Hence, it is clear that a longer time is required to cool the thick plastic parts compared to the thinner ones. All of these design cases provide significant improvement from the traditional straight-drilled designs of injection molds. These also provide insight into a range of variables in which the mold designers can work.

It is notable here that, starting from DOE-4, the variation of cooling time decreases in DOE-5, and then again in DOE-6. This indicates that when designing a conformal cooling channel configuration for a thick plastic part, a very small change can cause an improvement in cooling time. While designing such channels, mold designers need to conduct detailed design of experiments with a wide range of design variables and a small gap between levels to find out the most effective design for their purpose.

On the other hand, when designing for comparatively thin plastic parts, it is easier to find the most effective configuration, as a small change causes a noteworthy difference in cooling time, and designers can work on a specific range of design variables.

DOE-7, DOE-8, and DOE-9 are designed for conical=shaped plastic parts. The best results are 12.70 s, 19.65 s, and 15.96 s, which are obtained in cases 7.1, 8.3, and 9.7, respectively. It is notable here that these cooling times are quite shorter than those in the cylindrical case. As such, designers may consider replacing cylindrical bodies, such as bottle caps, bowls, plastic containers, etc., with conical-shaped ones whenever possible. Moreover, using draft angle in the part design can also help reduce the cooling times, too. DOEs 4–9 show significantly lower values of variances compared to the previous DOEs (1, 2, and 3). As we have considered cooling times only, the variances for DOE 4–9 are 6.13 s, 5.13 s, 1.94 s, 4.98 s, 1.16 s, and 1.30 s. This indicates that the datasets are closer to each other in the latter DOEs compared to the previous ones, hence the rationale behind planning DOEs 4–9 on the basis of whether DOEs 1–3 are correct.

It should be noted that the channel design configuration that provides the minimum cooling time for a one-mm thick cylindrical plastic part is exactly the same for the one-mm thick conical-shaped part, too. This happens again for the 3.5 mm and six-mm thick parts also, indicating that the results are quite reliable, and can be adopted for other shapes of plastic parts such as rectangular boxes or spherical balls. Figures 7–9 show the cooling time trends in different design cases. They provide a comparison of cooling times for similar cylindrical and conical shape designs for plastic bodies. For the same thickness, and with the same channel design, the conical parts are cooled down faster than the cylindrical designs. In Figure 10, the temperature distribution of the plastic parts for the best design cases as obtained in DOE 4, 5, 6, 7, 8 and 9 are shown.

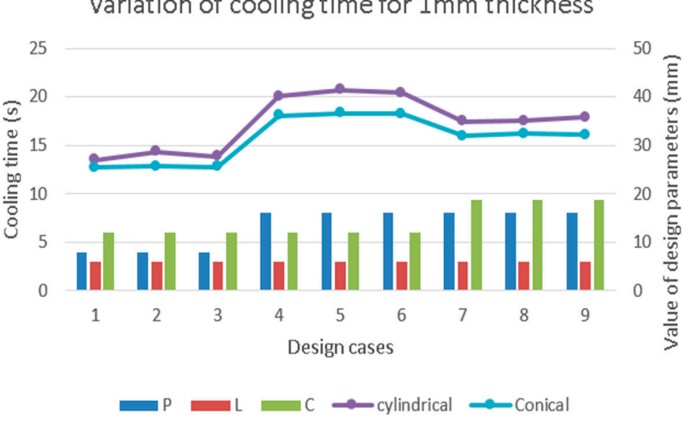

**Figure 7.** Cooling time variation for one-mm thick plastic parts (both cylindrical and conical).

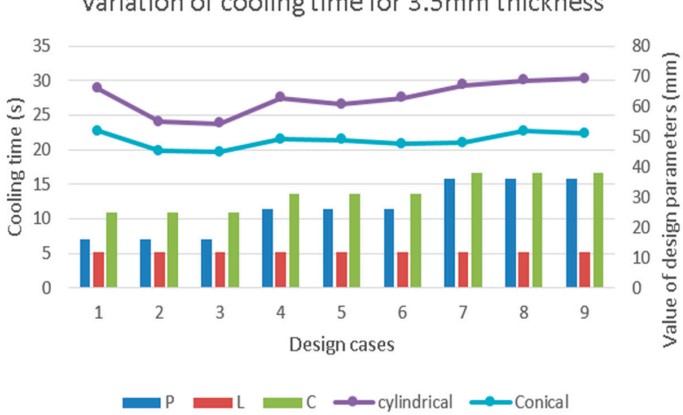

**Figure 8.** Cooling time variation for 3.5-mm thick plastic parts (both cylindrical and conical).

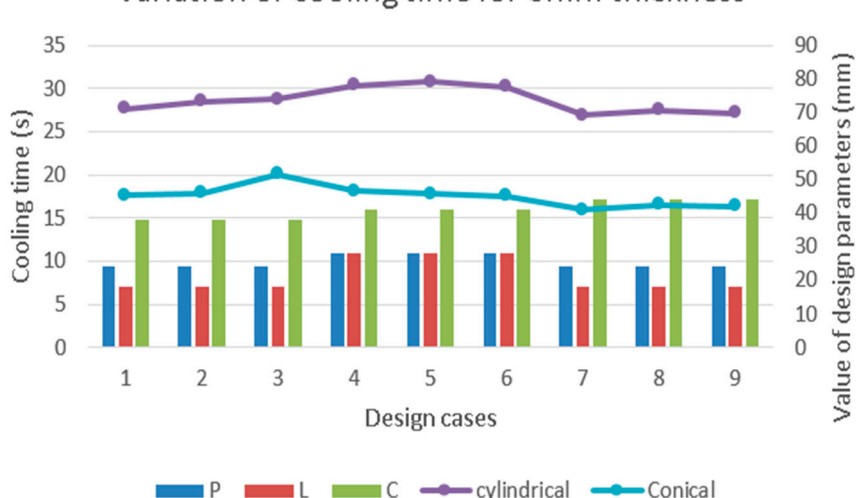

**Figure 9.** Cooling time variation for six-mm thick plastic parts (both cylindrical and conical).

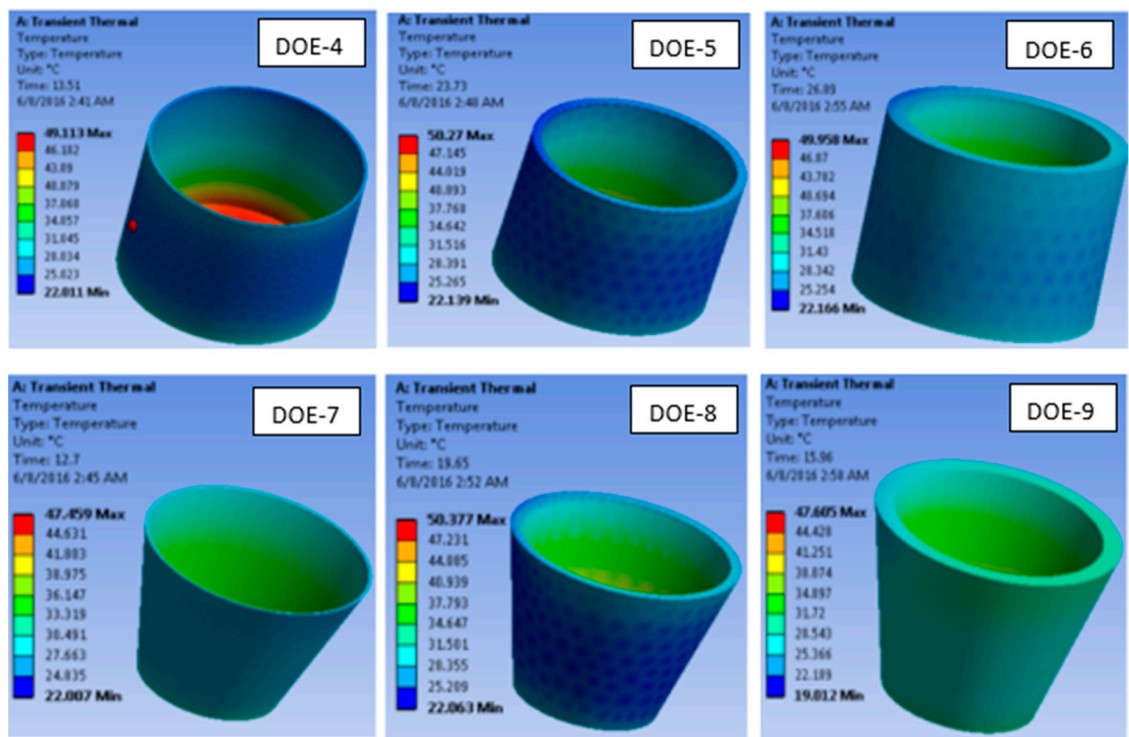

**Figure 10.** Temperature distribution on plastic parts for optimal design cases in DOE 4, 5, 6, 7, 8 and 9.

This study provides a basis and guidelines for mold makers to support their design conformal cooling channels for their injection molds. Due to design constraints in injection molds, such as the position of inlet and outlet ports, the gate, runner, ejector pins, etc., the best design solution may not be feasible in practical industrial cases. For this reason, the top three design cases are enlisted in Figure 11 for each size and shape of plastic parts, in order to provide more options to mold designers in the industry.

| Plastic Part Design | Conformal Channel Design Parameters | | |
|---|---|---|---|
| | Design Solution 1 (optimum) | Design Solution 2 (less optimal than Solution 1) | Design Solution 3 (less optimal than Solution 2) |
| Cylindrical 1mm | X section: circular, D=4mm, P=8mm, L=6mm | X section: square, 3.1 x3.1(mm x mm), P=8mm, L=6mm | X section: rectangular, 3.8 x2.5(mm x mm), P=8mm, L=6mm |
| Cylindrical 1.5mm | X section: rectangular, 5.6 x3.8(mm x mm), P=8mm, L=7mm | X section: circular, D=6mm, P=8mm, L=7mm | X section: square, 4.7 x4.7(mm x mm), P=8mm, L=7mm |
| Cylindrical 3.5mm | X section: square, 6.3 x6.3(mm x mm), P=16mm, L=12mm | X section: circular, D=8mm, P=16mm, L=12mm | X section: rectangular, 7.5 x5(mm x mm), P=16mm, L=12mm |
| Cylindrical 6mm | X section: circular, D=14mm, P=24mm, L=18mm | X section: rectangular, 13.2x8.8(mm x mm), P=24mm, L=18mm | X section: circular, D=12mm, P=24mm, L=18mm |
| Conical 1mm | X section: circular, D=4mm, P=8mm, L=6mm | X section: square, 3.1 x3.1(mm x mm), P=8mm, L=6mm | X section: rectangular, 3.8 x2.5(mm x mm), P=8mm, L=6mm |
| Conical 3.5mm | X section: square, 6.3 x6.3(mm x mm), P=16mm, L=12mm | X section: rectangular, 7.5 x5(mm x mm), P=16mm, L=12mm | X section: square, 7.9 x7.9(mm x mm), P=26mm, L=12mm |
| Conical 6mm | X section: circular, D=14mm, P=24mm, L=18mm | X section: square, 10.9 x10.9(mm x mm), P=24mm, L=18mm | X section: rectangular,13.2x8.8(mm x mm), P=24mm, L=18mm |

**Figure 11.** Conformal cooling channel design guidelines.

In addition to the structural stability of the mold and cooling time and production cycle time of the process, the plastic part quality is also important for the plastic injection molding industry. Hence, warping in the final plastic part is a crucial factor to determine the design of the cooling channels in the mold. A detailed study on the effect of warping can be found in the literature [46]. As discussed earlier, the traditional molds have straight drilled holes and buffers in some cases to enhance the cooling inside the mold, which cannot reach very close to the mold walls. On the contrary, "conformal" cooling channels 'conform' to the shape of the mold wall, and are capable of being very near the wall. As a result, the cooling becomes very uniform with such molds, and the warping also minimizes. In this study, warping was not included in the design consideration, as it would elongate the DOEs beyond its scope. In the future, we would incorporate the warping and demonstrate new DOE cases with the study.

## 4. Conclusions

In this study, a design methodology has been implemented to determine the optimum design of conformal cooling channels in injection molds. With the increase of competition in business, mold makers need efficient design tools to serve their purpose. This work provides guidelines to support such a need. A numerical model is developed to analyze the thermal and structural performance of injection molding tools. This model provides a base to compare the performance of various mold designs and conformal channel configurations. A number of design of experiments have been undertaken to identify the most suitable design of channels in injection molds. The results show that for different plastic part designs, different channel configurations provide optimum solutions in terms of the cross-section dimensions, section size, pitch distance, mold wall to channel centerline distance, etc. Hence, a guideline chart is provided in this study to help the mold designers choose the design parameters for their respective cases.

**Author Contributions:** This research article is a combined effort of S.A.J. and her research advisor H.E.-M. Conceptualization, supervision, funding acquisition, review and editing in contributed by H.E.-M., and methodology, software, validation, formal analysis, investigation, resources, data curation, writing, original draft preparation, visualization by S.A.J.

**Funding:** This research was funded by The Walmart Foundation.

**Acknowledgments:** The Walmart Foundation supported this research effort through the Walmart U.S. Manufacturing Innovation Fund. Any opinions, findings, conclusions, and recommendations expressed in this investigation are those of the writers and do not necessarily reflect the views of the sponsors.

**Conflicts of Interest:** The authors declare no conflict of interest. The funders had no role in the design of the study; in the collection, analyses, or interpretation of data; in the writing of the manuscript, or in the decision to publish the results.

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
