# Peer review of "A Thermomechanical Analysis of Conformal Cooling Channels in 3D Printed Plastic Injection Molds"

_applsci, doi:10.3390/app8122567_

Reviewer 1 Report

“A Thermo-Mechanical Analysis of Conformal Cooling 3 Channels in 3D printed plastic Injection molds” proposes a design methodology to generate optimized design configurations of conformal channels in plastic injection molds.

The work concerns an interesting aspect of injection moulding process and the adopted methodology should be able to give reliable suggestions on the studied aspects but:

Major revision:

The mixed level factorial designs were performed without an analyses of the results. Do runs (DOE 1,2,3) that were individuated because minimize the chosen quality indices give statistically valid results? With which confidence level?…

line 221, which is the optimization method adopted?

line 220, the simulation technique should be indicated and then referred.

Main effects plots and Pareto charts could resume and make clearer results.

The same difficulties are present for the other experiments (DOE 4-9)

Minor revision :

line 146-147 are not clear.

At the end of the introduction (line 152-154) a better resume of the forward sections is required.

Section are not correctly named. 3 could be Results and Discussion, 4 Conclusions

Author Response

Response to Reviewer 1 Comments 

Point 1: “A Thermo-Mechanical Analysis of Conformal Cooling 3 Channels in 3D printed plastic Injection molds” proposes a design methodology to generate optimized design configurations of conformal channels in plastic injection molds. The work concerns an interesting aspect of injection moulding process and the adopted methodology should be able to give reliable suggestions on the studied aspects. 

Response 1: Thanks for understanding the scope of the manuscript and considering the adopted methodology to be able to provide reliable results. 

 MAJOR REVISION 

Point 2: The mixed level factorial designs were performed without an analysis of the results. Do runs (DOE 1,2,3) that were individuated because minimize the chosen quality indices give statistically valid results? With which confidence level?…

Response 2: We have performed full factorial design of experiments in the study. There were mixed levels in determining the variables on the experiments. DOE-1,2,3 were performed for three different designs of plastic parts. In designing the tooling for plastic mold injection process, the final plastic product thickness is the determinant of the design of cooling channels in the mold. Hence, DOE 1,2 and 3 are for three different plastic part thicknesses. They are not mutually related, and such DOEs are hence more practical, rather than making all them mutually related. The results of these DOEs (1,2,3) are then further used in designing the latter DOEs. This is to obtain more optimum design results for cooling channels and reduce the computational time. We are not conducting any statistical analysis, but conducting thermo -mechanical simulations in different case studies to identify the most effective configuration. 

Point 3: line 221, which is the optimization method adopted?

Response 3: We have used Full factorial Design of Optimization method in this study, for all the DOEs. This is mentioned in the revised manuscript (line #212-214). Thanks for pointing this out. 

Point 4: line 220, the simulation technique should be indicated and then referred.

Response 4: We have used both transient thermal analysis and static structural analysis in this study. This is elaborated in the revision (line #224-235). The simulation technique is explained in detail in our previous publication and it is already mentioned and cited in the original submission. 

Point 5: Main effects plots and Pareto charts could resume and make clearer results. 

Response 5:  As mentioned earlier in response 2, we have not done any statistical analysis. The Design of experiments are based on thermo-mechanical results obtained at each case, and optimization is attained by analyzing those results. The effect of design variables on the cooling time and maximum stress is demonstrated in Figure 3, 4 and 5. Moreover, for the latter DOE cases, the effect on cooling time is presented in Figure 7, 8 and 9. 

Point 6: The same difficulties are present for the other experiments (DOE 4-9). 

Response 6: This is addressed in the revised manuscript. As mentioned in the response 2, 3 and 4, we have used full factorial DOE technique and the simulation parameters are described in the revision. 

MINOR REVISION 

Point 7: line 146-147 are not clear. 

Response 7: This issue is taken care of in the revision (line #148-150). 

Point 8: At the end of the introduction (line 152-154) a better resume of the forward sections is required. 

Response 8: Thank you for raising this concern. It is addressed in the revision (line# 157-160). 

Point 9: Section are not correctly named. 3 could be Results and Discussion, 4 Conclusions

Response 9: We completely agree with your comment. It is modified in the revised submission (line #257 and line #451).

Reviewer 2 Report

The paper discusses a topic that is highly relevant in today’s manufacturing and provides useful insight to mould designers and plastic processors. 

The literature study is comprehensive and set the basis for the motivation behind the article. The conclusion is supported by the results. 

The paper is inherently of a technical nature, since the scientific findings behind the key enabling technologies supporting the study (AM, simulations, etc) are very well known and part of the current state of the art. 

To make the article fully relevant for the industry community, the authors should consider adding another output to their DOE studies: the warpage of the parts as a function of the different cooling designs. They should also define the plastic material used for the injection moulding process so that it is possible to understand the level of key process parameters employed such as the injection pressure applied, the melt temperature, the mould temperature, that are crucial for cycle time and warpage. 

It is of course very important to minimise cooling time and the max stress, but at the same time it is equally important to maximise the quality of the produced parts, and in Injection Moulding this is enabled by the design of the cooling system. As a consequence the cooling system design must deal with both optimisations: minimisation of cooling time and of warpage. 

An example of the influence of cooling system design on the part warpage can be found on this publication: http://orbit.dtu.dk/files/55911264/Performance_evaluation.pdf

Author Response

Response to Reviewer 2 Comments

Point 1: The paper discusses a topic that is highly relevant in today’s manufacturing and provides useful insight to mould designers and plastic processors. The literature study is comprehensive and set the basis for the motivation behind the article. The conclusion is supported by the results. The paper is inherently of a technical nature, since the scientific findings behind the key enabling technologies supporting the study (AM, simulations, etc) are very well known and part of the current state of the art. 

Response 1: Thanks for understanding the scope and outcomes discussed in the manuscript. This is of course a technical paper and we are trying to establish state of the art design methods and techniques for mold designers and enable implementation of available manufacturing technologies.

Point 2: To make the article fully relevant for the industry community, the authors should consider adding another output to their DOE studies: the warpage of the parts as a function of the different cooling designs. They should also define the plastic material used for the injection moulding process so that it is possible to understand the level of key process parameters employed such as the injection pressure applied, the melt temperature, the mould temperature, that are crucial for cycle time and warpage.

Response 2: This is true that, warpage of the parts is very important for plastic injection industry, is it is a determining factor of the product quality. But it is difficult to incorporate the warpage study for all the DOE cases described in the paper, is it will elongate the paper beyond its scope. But, as the deigns of conformal cooling provides faster and uniform cooling, the warpage on the parts are also minimized. Compared to straight drilled cooling channels (which is the existing build feature of traditional molds), any design of conformal cooling channel provides more uniform cooling on the mold as well as on the plastic part (as shown in Figure 2). Hence in this paper, we have considered the effect of cooling time and stress distribution as determining factor of optimization.

Regarding the plastic material used in the injection moulding process, it is mentioned in the previous publications of the research group and cited in the manuscript (reference# 35,40 in original manuscript). Yet, to clarify, we have added these in formation in the revised version (line #224-235).

Point 3:  It is of course very important to minimise cooling time and the max stress, but at the same time it is equally important to maximise the quality of the produced parts, and in Injection Moulding this is enabled by the design of the cooling system. As a consequence the cooling system design must deal with both optimisations: minimisation of cooling time and of warpage. 

Response 3:  As discussed in the previous response, including the effects on warping was beyond the scope of discussion in this manuscript. But, in our future publications, we will incorporate the effects of warping, surface roughness and few more important characteristics of the final plastic product in the DOE study. Thanks for pointing this out. We have included a discussion on warpage and what would be the effect of cooling channel on the warpage of plastic product in the revised version of the manuscript (line # 437-447). 

Point 4: An example of the influence of cooling system design on the part warpage can be found on this publication: http://orbit.dtu.dk/files/55911264/Performance_evaluation.pdf.

Response 4: Thank you for this reference. This is very useful. This reference is also cited in our revision. (line#440).

Reviewer 3 Report

The paper deals with  a methodology to design configurations of  channels in plastic injection molds. The paper is well written and is of interest to the community. I recommend publication.

Author Response

Response to Reviewer 3 Comments

Point 1: The paper deals with a methodology to design configurations of channels in plastic injection molds. The paper is well written and is of interest to the community. I recommend publication.

Response 1: Thank you very much for your review and comments. We really appreciate it, and looking forward to publishing as soon as possible.

Reviewer 4 Report

The article investigated a design methodology to generate optimized design configurations of conformal cooling channels in plastic injection molds. The article is well written and results are good. 

Some suggestion are given below:

1. Specify which kind of Design of Experiment was used (Response Surface, Full factorial, etc.)

2. Pleas add the Analysis of Variance foe each DoE plan.

3. Please add the following references:

- Spina R., Spekowius M,  Hopmann C., Multiphysics simulation of thermoplastic polymer crystallization, Mat. Des. 95 (2016) 455-469. --> add after line 34 --> research pointed out the effect of the process parameters on polymer crystallization.

- Zink B,  Kovács FG, The effect of limescale on heat transfer in injection molding, Int. Commun. Heat Mass Transfer 86 (2017) 101-107. add after line 111 --> Research pointed out out limescale on cooling efficiency, also  in case of conformal cooling channel. 

Author Response

Response to Reviewer 4 Comments

Point 1: The article investigated a design methodology to generate optimized design configurations of conformal cooling channels in plastic injection molds. The article is well written, and results are good.

Response 1: Thank you for your comment. We really appreciate it.

Point 2: Some suggestions are given below:

1. Specify which kind of Design of Experiment was used (Response Surface, Full factorial, etc.)

Response 2: We have used Full factorial method in the Design of experiment. This is mentioned in the revised manuscript (line# 212-214). Thanks for pointing it out.

Point 3:  2. Please add the Analysis of Variance foe each DoE plan.

Response 3: The variance analysis is added for each DOE in the revised manuscript (line # 315-318, line #407-410).

Point 4: 3. Please add the following references:

- Spina R., Spekowius M,  Hopmann C., Multiphysics simulation of thermoplastic polymer crystallization, Mat. Des. 95 (2016) 455-469. --> add after line 34 --> research pointed out the effect of the process parameters on polymer crystallization.

- Zink B,  Kovács FG, The effect of limescale on heat transfer in injection molding, Int. Commun. Heat Mass Transfer 86 (2017) 101-107. add after line 111 --> Research pointed out out limescale on cooling efficiency, also  in case of conformal cooling channel.

Response 4: Thank you for these references. These are very useful and informative. We have added them in our revised manuscript (line # 34-35 and line# 112-114).

Round  2

Reviewer 1 Report

The present form improves the work and make clear results